# Size and Number of Food Boluses in the Stomach after Eating Different Meals: Magnetic Resonance Imaging Insights in Healthy Humans

**DOI:** 10.3390/nu13103626

**Published:** 2021-10-16

**Authors:** Hannah Hornby, Mar Collado-González, Xue Zhang, Nichola Abrehart, Meshari Alshammari, Serafim Bakalis, Alan Mackie, Luca Marciani

**Affiliations:** 1NIHR Nottingham Biomedical Research Centre, Nottingham University Hospitals NHS Trust and Translational Medical Sciences, School of Medicine, University of Nottingham, Nottingham NG7 2UH, UK; mzyheh@exmail.nottingham.ac.uk (H.H.); Nichola.Abrehart@nottingham.ac.uk (N.A.); mzxma11@exmail.nottingham.ac.uk (M.A.); 2School of Food Science and Nutrition, University of Leeds, Leeds LS2 9JT, UK; M.D.M.ColladoGonzalez@leeds.ac.uk (M.C.-G.); foodimaging123@gmail.com (X.Z.); 3Department of Food Science, University of Copenhagen, DK-1958 Copenhagen, Denmark; bakalis@food.ku.dk

**Keywords:** intragastric, in vivo, digestion, distribution, MRI, meal composition, bolus size

## Abstract

Oral processing of food results in the formation of food boluses, which are then swallowed and reach the stomach for further digestion. The number, size and surface properties of the boluses will affect their processing and emptying from the stomach. Knowledge of these parameters, however, is incomplete due to limitations of the techniques used. In this work, non-invasive magnetic resonance imaging (MRI) was used for the first time to measure boluses in the stomach a few minutes after swallowing. Three groups of nine healthy participants were fed three different meals: chicken and roasted vegetables (Meal 1), bread and jam (Meal 2) and cheese and yogurt (Meal 3), and then, their stomach content was imaged. The median number of boluses within the stomach was 282, 106 and 9 for Meal 1, Meal 2 and Meal 3 (*p* < 0.0001) with an average volume of 0.47 mL, 2.4 mL and 13.6 mL, respectively (*p* < 0.0001). The cohesiveness as well as the meal composition seem to play a key role in the resulting boluses. These new in vivo data from undisturbed organ imaging can improve knowledge of the digestion process, which will, in turn, inform in vitro and in silico modelling of digestion, thus improving their in vitro/in vivo relevance.

## 1. Introduction

Digestion comprises a series of sequential processes, whereby each stage of digestion is influenced by the previous one. In particular, the properties of the bolus formed in the mouth can influence gastric behaviour, and thus, digestion kinetics and consequent physiological outcomes. Oral bolus properties are a function of the properties of the food consumed, including its structure, texture and composition. It is then influenced by individual physiology, mastication patterns and saliva [1].

Digestion starts in the mouth, where the main goal is to reduce the size and increase the lubricity of the food particles before swallowing. Mastication incorporates saliva to increase the lubrication and cohesiveness of the food particles [2]. Salivary amylase starts the hydrolysis of starch in the mouth, and the movements of the tongue mix food particles and assist the comminution and swallowing processes [3] that play important roles in bolus formation within this first part of digestion [4,5]. Food properties such as hardness, texture, initial moisture, and fat content will affect the bolus properties [6,7].

The texture and elasticity of food as well as the particle size of food in the mouth regulate the muscle force applied in chewing food, the volume of saliva secreted and the number of cycles of mastication, resulting in different final bolus properties [8]. Research comparing the oral digestion of nuts and raw vegetables showed that vegetables, on average, resulted in larger particle size in the bolus compared to nuts [9]. There are known criteria that a food bolus needs to meet before it can be swallowed [10]. These include factors such as bolus cohesion and elasticity [11]. Other important factors include the mastication pattern and the duration and number of cycles of chewing, which can vary between subjects [9].

Food boluses are not rigid structures. Their shape is dynamic and continually changing on their journey through the mouth to the oesophagus and into the stomach [4,5,12]. Boluses are irregular in shape in the mouth and they become elongated during their passage through the oesophagus thanks to their cohesiveness. During swallowing and on arrival into the proximal part of the stomach, limited mixing takes place but salivary amylase begins to digest the starch in the bolus [5]. The bolus then moves through the stomach mixing with gastric acid secretion to form chyme. In the final stage of gastric digestion, the bolus is subjected to peristaltic muscle contractions that vary in frequency and intensity to achieve a reduction in particle size and subsequent passage through the pylorus [12] into the small bowel [5]. In the duodenum, there are receptors to detect the pH and nutritional content of the chyme. Stomach content, hormone release and the position of the chyme within the stomach drive the contractions and peristalsis [13]. Solids are subjected to both chemical and physical digestion before gastric emptying can occur [5,14]. In a clinical research study, the difference in gastric emptying rate was recorded between solids with different sizes. This study reported that the half emptying time for 1 mL chicken liver cubes was 75 min, while the half emptying time for 0.027 mL chicken liver cubes was 50 min [15].

When two food phases are present, solid and liquid, gastric sieving can take place [13,16]. This drives a phase separation of liquids from solid foods, leading to a faster gastric emptying of the liquid phase [17]. Moreover, gastric digestion is affected by the consistency, structure and size of the bolus. The acidic content of the stomach will act faster on less dense boluses that will also require fewer peristalsis movements than more dense or compacted boluses [18]. Once the solid phase particles reach the size that allows them to pass to the duodenum, usually reported to be between one and a few millimetres in diameter, the rate of gastric emptying is the same as that shown by the equivalent liquid phase [19].

The transformation of food materials in the gastrointestinal tract has attracted significant interest. The formation and consequent break up of food boluses can lead to different digestibility. Understanding and eventually controlling the structure of food boluses will enable better modelling and design of foods with controlled delivery of nutrients. This knowledge would be of interest to the wider food and nutrition communities. There is also an increasing interest in the community in designing in vitro methods that would capitalize knowledge to limit the need for animal and human trials. Food bolus properties have been studied before using different techniques, such as a series of sieves, laser diffraction and image analysis. All these processes require the subjects to sham-feed and spit out the bolus when they felt the impulse to swallow [4]. Although these techniques allow good characterization of bolus content, the swallowing and oesophageal transit processes are missing; hence, the sham-fed bolus size may not entirely reflect what would arrive in the stomach.

In this study, the size of the swallowed bolus was investigated in vivo using Magnetic Resonance Imaging (MRI), which has the advantages of being both non-ionizing and a non-invasive technique. MRI allows imaging of the inside of the stomach in real time without disturbing or interrupting the process. MRI has been used previously to analyse the appearance and total volume of food content in the stomach [20,21]. In this work, T2-weighted scans were used. This particular type of MRI imaging highlights the signal from water-rich areas that appear as bright regions, whilst areas with lower water mobility such as muscles, liver and solids appear as dark regions within the stomach.

We hypothesized that MRI imaging would allow visualization of the individual food boluses in the stomach and that this would provide new knowledge on food bolus size and distribution after they have been swallowed. Therefore, this study aimed to measure, for the first time, the volume and number of food boluses found in the stomach of healthy volunteers. MRI scans taken from three previous studies in healthy participants who were fed three markedly different type of meals were used.

## 2. Materials and Methods

### 2.1. Study Design

This was a retrospective study. Nine image datasets were selected at random, with no specific biometric selection criteria, from each group of participants studied in three previous studies that addressed gastric emptying of three different meals but did not consider or measure intragastric food bolus size. The first study [16] recruited 18 healthy participants, the second study [22] recruited 12 and the third study [23] recruited 10. The first image time point taken after feeding was chosen for each participant and analysed as described below. Considering the time allowed for the participants to consume their meals and the time of the first MRI scan, the images were taken at comparable time points after meal intake across the three studies, which was approximately 15 min after starting to consume the meals.

### 2.2. Test Meals

Throughout the manuscript, the three meals will be identified as:

Meal 1: the data came from Marciani et al. (2012) [16]. Meal 1 consisted of 75 g chargrilled chicken (Tesco, Nottingham, UK), 62.5 g roasted vegetables containing pepper, courgette, red onion and cherry tomato (Tesco “finest” Mediterranean style vegetables, Tesco, Nottingham, UK) and 62.5 g breaded mushrooms (Tesco “value” breaded mushrooms, Tesco, Nottingham, UK). Meal 1 was consumed with 250 mL of bottled still water. Total portion size = 450 g.

Meal 2: the data came from Coletta et al. (2016) [22]. Meal 2 comprised of 150 g (approximately 4 slices) of white bread manufactured by Campden BRI using commercial flour and methods. The bread was served with 24 g margarine and 34 g seedless raspberry jam (Sainsbury’s, Nottingham, UK). The meal was consumed with 100 mL pure orange juice from concentrate (Sainsbury’s, Nottingham, UK). Total portion size = 308 g.

Meal 3: the data came from Mackie et al. (2013) [23]. Meal 3 consisted of 88 g of finely grated Gouda cheese (Waitrose Essential Dutch Gouda, Waitrose, Leeds, UK) and 73 g of low-fat yogurt (Waitrose Essential low-fat yogurt, Waitrose, Leeds, UK). The meal was consumed with 339 mL of bottled still water. Total portion size = 500 g.

The nutritional composition of the three meals used is given in Table 1.

### 2.3. Participants

All three studies enrolled healthy adult participants. For Meal 1: 3 males and 6 females, mean age 20.1 years and mean BMI 21.9 kg/m^2^. For Meal 2: 2 males and 7 females, mean age 26.7 years and mean BMI 22.7 kg/m^2^. For Meal 3: 9 males and no females, mean age 35 years and mean BMI 24.7 kg/m^2^. The participants were asked to fast before ingestion of the respective test meals, an overnight fast for Meal 1 and Meal 2, and a 5 h fast for Meal 3.

### 2.4. MRI Imaging

Participants were positioned supine in the MRI scanner with a parallel imaging receiver coil wrapped around the abdomen. For Meal 1 and Meal 2, imaging was carried out on a 1.5 T Philips Achieva scanner (Philips, Best, The Netherlands). For Meal 3, a 1.5-T Siemens Avanto scanner (Siemens, Erlangen, Germany) was used. The participants spent approximately 5–10 min inside the magnet for the acquisition including set up. For Meal 1 and Meal 2, the stomach was imaged using a balanced gradient echo (balanced turbo field echo) sequence, and the data were collected during an expiration breath hold of approximately 10 s, monitored using a respiratory belt. For Meal 1, 20 contiguous axial slices were acquired with a reconstructed in-plane resolution of 1.56 mm × 1.56 mm, repetition time (TR)/echo time (TE) 2.4/1.2 ms; field of view 40 × 32 cm; reconstructed matrix 256 × 256; slice thickness 1 cm. For Meal 2, 25 contiguous axial slices were acquired with a resolution of 2.01 mm × 1.76 mm; TR/ TE 2.8/1.4 ms; field of view 40 × 32 cm; reconstructed matrix 256 × 256; slice thickness 1 cm. For Meal 3 scans, a TRUFISP (fast imaging with steady-state precession) sequence was used using a breath-hold of 15–25 s depending on the fullness of the stomach, TR/TE 3.5/1.5 ms; field of view 24 × 32 cm; matrix 154 × 256; slice thickness 0.5 cm.

### 2.5. Data Analysis

Data analysis was carried out using ImageJ Version 1.8.0_172 (NIH, Bethesda, MD, USA). A food bolus was defined here as an individual volume of food in the stomach that was distinguishable in the images from the surrounding liquid or chime. This could have been a solid particle or a more cohesive semi-solid mass. In the MRI images, a bolus appeared as an area inside the stomach that was darker than the surrounding liquid or chyme and had distinct edges, allowing it to be segmented. Meal 1 and Meal 2 were analysed by one trained observer and Meal 3 was analysed by a second trained observer, using a standardized method. Both observers re-assessed a subset of image data and no significant intra-operator variability was found. Prior to taking measurements, the correct scale to match the MRI scanner resolution was set. The stomach was identified on each slice of the MRI scans and each food bolus identified was manually drawn around using the polygon selection tool. Measurements were taken for area, centre of mass (which gave coordinates) and stack position for each bolus. Care was taken to check if the bolus spanned several slices and if this was the case, the areas were added together to be calculated as one bolus. These measurements were then transferred onto an Excel spreadsheet and all areas multiplied by the slice thickness to yield the volume of the bolus. Having measured all boluses, volume values underwent further analysis using GraphPad Prism Version 9.0.0 (GraphPad Software, San Diego, CA, USA).

The output parameters considered included total number of boluses counted, total volume of food boluses in the stomach (mL), mean food bolus volume measured (mL), mean percentage ratio of food bolus within the stomach by using Equation (1), total energy per mL of food bolus in the stomach by using Equation (2), and surface area of each food bolus (cm^2^). The surface area of the food boluses was estimated assuming that the food boluses all have spherical shape. With the data for each food bolus volume, the surface area of a sphere with the same volume was then simply calculated. All calculated surface area measurements were then added together to derive the value for total surface area of all food boluses in the stomach of each participant. Qualitative observations describing the look and overall appearance of the food in the stomach were also collected.
(1)$$Ratio\;of\;food\;bolus=\frac{total_{bolus\;volume}}{total_{stomach\;content}}\cdot 100$$
(2)$$Energy\;per\;mL\;of\;bolus=\frac{total\;energy_{\;meal}}{total_{bolus\;volume}}$$

### 2.6. Statistical Analysis

Many datasets were not normally distributed and the data are presented either as individual values or as median (interquartile range IQR). Statistical analysis was carried out on datasets using GraphPad Prism Version 9.0.0 (GraphPad Software, San Diego, CA, USA). Firstly, Shapiro–Wilk’s test was used to establish if the data were normally distributed. If one or more of the datasets failed the normality test, then a non-parametric test was used to test the significance of differences. A *p* value of < 0.05 was considered significant. Intra-participant differences were analysed separately to determine significance of intra-participant differences using one-way ANOVA or Kruskal–Wallis test depending on the normality of the data. When the ANOVA indicated a statistically significant difference, post hoc analysis between each group and every other group was carried out using Dunn’s multiple comparison test.

## 3. Results

### 3.1. Intragastric Appearance

Good-quality images were obtained from all participants. There was a variety of different visual appearances, types and shapes of food boluses in the stomach. Figure 1 shows an example for each meal. In this type of MRI image weighting, fluids in the stomach appear bright and solids appear dark. Meal 1 showed a distribution dispersed in the stomach, comprising principally smaller boluses surrounded by fluid, but also larger boluses of heterogeneous appearance. These appeared to be clumps containing multiple smaller boluses, possibly containing different components of this chicken and vegetable mixed meal, formed in the mouth and upon swallowing. These larger boluses appeared mostly in the proximal stomach. Gravity seemed to play a part too with the smaller boluses often seen at the top of the stomach. More homogeneous chyme of a brighter appearance, suggesting increased hydration, could be seen in the distal antral region for Meal 2, whilst separate or sometimes clumped together bread boluses were located more proximally in the body region of the stomach. These were larger boluses with a lower amount of surrounding fluid. Finally, the content of the stomach of those who ingested Meal 3 resembled a phase separation. Large boluses were seen located close to each other and in the bottom part of the stomach due to the gravity effect, noting that participants were in supine position.

### 3.2. Frequency Distribution of Food Boluses

Figure 2 shows the frequency distribution of the number of food boluses integrated between all nine participants for each meal. The size bins were spaced logarithmically with the number of 27 bins determined from the number of observations using Sturge’s rule. The distributions for Meal 1 and Meal 2 look similar, with an average modal volume range for all participants of 0.1–0.19 mL for both these meals. This shows that the majority of boluses were at the smaller volume ranges; however, Meal 1 has a higher frequency of very small boluses up to 0.1 mL, whilst Meal 2 peaks around 0.3 mL in size with occurrences also at the larger volume ranges between 3.3 and >100 mL. When calculated, a strong positive skew of 2.77 was found for Meal 1 and of 3.15 for Meal 2. Meal 3 has a markedly different distribution, with the majority of boluses counted being at the higher ranges of volume. The modal range is between 6 and 7.99 mL and a negative skew of 1.50 was calculated, in contrast with both Meals 1 and 2. The differences in frequency counts between the meals are significantly different (Kruskal–Wallis test *p* < 0.0001).

### 3.3. Number of Food Boluses

For each meal, the number of food boluses was calculated for each participant and the data are shown in Figure 3A. The number of food boluses was markedly different for the different meals with the median (IQR) values being 282 (198–339) boluses for Meal 1, 106 (84–151) boluses for Meal 2 and 9 (6–13) boluses for Meal 3. The difference in number of food boluses was significantly different between meals (Kruskal–Wallis test *p* < 0.0001). Post hoc analysis showed a significant difference between Meal 1 versus Meal 2 (Dunn’s multiple comparison test *p* < 0.05), Meal 1 versus Meal 3 (Dunn’s multiple comparison test *p* < 0.0001) and Meal 2 versus Meal 3 (Dunn’s multiple comparison test *p* < 0.05).

### 3.4. Volume of Food Boluses

For each meal, the average volume of food boluses was calculated for each participant and the data are shown in Figure 3B. The volumes of food boluses were markedly different for the different meals with the median (IQR) values being 0.47 (0.43–0.63) mL for Meal 1, 2.4 (2.0–3.6) mL for Meal 2 and 13.60 (7.8–17.4) mL for Meal 3. The difference in average volume of food boluses was significantly different between meals (Kruskal–Wallis test *p* < 0.0001). Post hoc analysis showed a significant difference between Meal 1 versus Meal 2 (Dunn’s multiple comparison test *p* < 0.05) and Meal 1 versus Meal 3 (Dunn’s multiple comparison test *p* < 0.0001). For Meal 1, the smallest bolus measured was 0.02 mL and the largest 12.4 mL. For Meal 2, these were, respectively, 0.05 and 161 mL and for Meal 3, respectively 0.3 and 119 mL.

For each participant, the total volume of food boluses present in the stomach was then calculated by integrating each food bolus volume measured by subject. The data are shown in Figure 3C. The total volume of food in the stomach was markedly different for the different meals with the median (IQR) values being 136 (129–165) mL for Meal 1, 297 (262–312) mL for Meal 2 and 118 (83–125) mL for Meal 3. The difference in total volume of food in the stomach was significantly different between meals (Kruskal–Wallis test *p* < 0.0001). Post hoc analysis showed a significant difference between Meal 1 versus Meal 2 (Dunn’s multiple comparison test *p* < 0.05) and between Meal 2 versus Meal 3 (Dunn’s multiple comparison test *p* < 0.0001).

The values of total volume of gastric contents were available from the original work so it was possible to calculate the ratio of boluses volume to total gastric contents, which include liquid and also very small fragments. This is shown in Figure 3D. The ratios were, respectively, 33 (26–38) for Meal 1, 54 (52–66) for Meal 2 and 20 (16–23) for Meal 3, with differences being statistically significant (one-way ANOVA *p* < 0.0001). Post hoc analysis showed a significant difference between Meal 1 versus Meal 2 (Dunn’s multiple comparison test *p* < 0.05) and between Meal 2 versus Meal 3 (Dunn’s multiple comparison test *p* < 0.0001).

Furthermore, from the total boluses volume and the data in Table 1, it is possible to calculate the average energy content per mL of food bolus in the stomach. The data are shown in Figure 3E. The average energy content per mL of food bolus in the stomach was, respectively, 1.8 (1.5–1.9) kcal/mL for Meal 1, 2.1 (2.0–2.3) kcal/mL for Meal 2 and 3.2 (2.0–4.5) kcal/mL for Meal 3, with differences being statistically significant (Kruskal–Wallis test *p* < 0.0001). Post hoc analysis showed a significant difference between Meal 1 versus Meal 3 (Dunn’s multiple comparison test *p* < 0.0001).

### 3.5. Total Surface Area of Food Boluses

Assuming that the food boluses all have spherical shape, it was possible to calculate the surface area for each bolus from their measured volume. All calculated surface area measurements were then added together to derive the total (cumulative) value for total surface area of all food boluses in the stomach of each participant. These data are shown in Figure 3F. The total surface area of food boluses in the stomach was markedly different for the different meals with the median (IQR) values being 713 (697–837) cm^2^ for Meal 1, 595 (518–758) cm^2^ for Meal 2 and 203 (142–237) cm^2^ for Meal 3. The difference in total surface area of food boluses was significantly different between meals (Kruskal–Wallis test *p* < 0.0001). Post hoc analysis showed a significant difference between Meal 1 versus Meal 3 (Dunn’s multiple comparison test *p* < 0.0001) and between Meal 2 versus Meal 3 (Dunn’s multiple comparison test *p* < 0.01).

### 3.6. Inter-Individual Variations

The data from each participant by meal showed marked and significant inter-individual differences in food bolus volume and surface area. These are plotted in Figure 4.

All individual data are provided in Appendix A.

## 4. Discussion

Magnetic resonance images of boluses within the stomach of 27 volunteers were used to quantitatively characterize boluses in the first stages of gastric digestion. Other techniques have been used to analyse gastric content, such as gamma scintigraphy and ultrasound techniques. However, both techniques have limitations. Gamma scintigraphy has low spatial resolution and images the radioactive label rather than the food. Ultrasound views are mostly limited to the antral region and the air–fluid interfaces typically present in the stomach affect image quality. The advantages of MRI are the ability to image the whole stomach, as well as its content, with high-resolution and without using ionizing radiation [24,25]. Once a complete two-dimensional stack of images of the stomach is acquired, 3D reconstruction and analysis of the volume of the boluses within the stomach are then possible using freely available image analysis software packages.

Since the structure of the food plays a key role in the digestion process, the meals included in this work were selected according to their physical properties. Namely, Meal 1 was a solid, Meal 2 was a soft solid and Meal 3 was a semisolid food. Additionally, given their macronutrient composition, comparison between different meals can be established since Meal 1 and Meal 2 are rich in fibre, Meal 2 is rich in carbohydrates and lipid, whereas Meal 3 is rich in protein and lipid.

The use of MRI to analyse the boluses is a new methodology and cannot be compared with previous reports. Methods that involve the chewing and expectoration of boluses (sham feeding) result in the loss of more than 50% of the initial mass of the food [9,26]. The use of MRI to analyse the food boluses does not involve losing mass, ensuring realistic results, particularly for the bolus volume.

Meal 3 had the least surface area/volume, resulting in less surface interaction with the surrounding gastric fluid. The surface area depends on the size of the particles and will vary because of both the intrinsic properties of the boluses and potential interaction (cohesion) among them after being swallowed and reaching the stomach.

The complete mechanical and chemical disintegration of boluses within the stomachs analysed here was incomplete. According to findings in the literature, the “lag” time (the period between the bolus arriving in the stomach and the gastric chyme entering the duodenum) has been reported as (62 ± 5) minutes for chicken liver [27]. The lag time can also be expected to vary as a function of the food properties including the bolus size. It has also been reported that 7.4 min after the ingestion of chicken particles (0.2–0.5 cm in diameter, 0.004–0.065 mL, assuming spherical particles), chyme was observed in the duodenum [28]. Although the bolus volumes reported in this work suggest a lag period longer than 7.4 min, the inter-individual variation in total bolus volume suggests that some emptying had started in some individuals. The time point at which the images were taken for the three meals was comparable and approximately 15 min after starting the meal.

It is highly probable that the water content of each meal plays a key role in the variability of the results. In a study analysing boluses made from bread, it was reported that the water content was increased from 40% to 65% due to the addition of saliva while chewing, resulting in a cohesive bolus [29,30]. Thus, the hydration factor could explain the total bolus volume and the greater bolus volume from the subjects that ingested Meal 2 that contained 72% in mass carbohydrates. The appearance of adhesion forces between bread boluses that results in the agglomeration of particles and ended with a cohesive mass has been previously proposed [31]. In addition, it has been previously reported that the hardness, adhesiveness and cohesiveness of food boluses are triggering factors for swallowing [6]. It has been proposed that food composition and structure are key factors that influence food bolus properties, with mastication of harder foods resulting in smaller boluses than those resulting from softer foods [32]. Such a result was found when comparing food boluses from Meal 1 and Meal 3, which have similar macronutrient composition. Despite the fact that this is the first study analysing the behaviour of food boluses within the stomach, some bolus characteristics are clear. The cohesiveness of the food boluses, as reported previously in non-swallowed food boluses [6,26,33], is likely to be an important factor in whether boluses coalesce into a single bolus or disaggregate into smaller particles. The influence of such particle interaction and their derived properties, such as bolus size and surface area, and their influence on gastric behaviour should be further studied. In summary, the results highlight the importance of properties such as cohesiveness in the volume of the boluses.

The association between gastric emptying and appetite means that it is of interest to investigate the effect of bolus characteristics on the rate of gastric emptying of the different meals. The half emptying times for the different meals were available from the previous studies as 78 (53–103) min for Meal 1, 151 (119–183) min for Meal 2 and 103 (52–154) min for Meal 3. As expected, the gastric emptying times showed a strong linear correlation with the total number of calories contained in the meals (r^2^ = 0.99) [34]. A comparison between Meals 1 and 2 also reveals that gastric emptying half time has a positive correlation (r^2^ = 0.6) with the initial size of the bolus particles. This implies that larger particles will take longer to break and eventually to be emptied from the gastric compartment. This is important as it would enable correlations to be drawn between the properties of the food (e.g., textural characteristics) and gastric emptying rate. Interestingly, this relationship breaks when Meal 3 is considered. The bolus size for this meal at the beginning of gastric digestion was substantially higher than that of Meals 1 and 2 (see Figure 3). However, gastric emptying was faster when compared to Meal 2. Meal 3 consisted of dairy proteins that have complex interactions with the gastric environment, (e.g., gelation and/or swelling), which govern gastric particle size. Furthermore, when Meal 3 was compared to the same meal in a liquid format, differences in gastric emptying were observed only for the first hour [23], indicating that the structure of the bolus can change significantly over time. The rate of gastric emptying is known to depend on a number of parameters, including caloric density, nutrient release in the small intestine, bolus structure, physiology, and hormonal release [34]. Nonetheless, this work shows that the initial size of the bolus may also affect the rate of gastric emptying.

One of the limitations of the study was the identification of food boluses in the stomach. A bolus (from the Latin bolus or “ball”) was defined here broadly as an individual volume of food in the stomach that was distinguishable in the images from the surrounding liquid or chime. This could have been a larger solid particle (e.g., a chunk of roasted chicken) or a more cohesive semi-solid mass (e.g., like those observed for Meal 3). A bolus could be identified as any area inside the stomach that was darker than the surrounding liquid or chyme and had distinct edges, enabling the operator to draw around it with the image analysis software. The segmentation of the boluses was manual, so this necessarily added an element of subjectivity to the analysis, although intra-operator repeatability was not a concern. Image contrast was not formally standardized, but it was kept relatively consistent by the operator for each participant and only adapted if a particular dataset needed it to make the contrast sharper. More watery components—for example, some of the vegetables in Meal 1—could have yielded lower contrast against fluid/chime, thus making the identification more difficult. There was also a possible issue with determination of boundaries of the boluses, as different boluses could appear as linked volumes. This could be explained in different ways, such as the merging of initial swallowed bolus parts, when the boluses reach the stomach, or the breaking of larger swallowed boluses into smaller ones when the mechanical digestion occurs. It is worth noting that Meal 2 was consumed with less fluid than the other meals, which could have affected identification of the contrast between boluses; although, from the images, this did not seem the case, and the body does add fed state secretion.

The methodology applied in this work has some other limitations. Selecting an equal number of nine participants from each of the three previous studies could have introduced some selection bias, but this risk was minimized by selecting the datasets randomly. There were some differences in demographic characteristics between studies; for example, for Meal 3, there were predominantly male participants. Such differences in group characteristics might have introduced some bias, for example, in chewing or eating behaviour. Conversely, this made the groups’ biometric characteristics such as age, gender and body mass index not ideally comparable. The three meals had different total portion size in weight and the original studies did not assess initial meal volume, which might have had an influence on the results. The food boluses were not perfect spheres and some had a shape that was far from spherical. The assumption of spherical shape was only made to enable an approximate calculation of surface area of the food boluses to enable initial inferences on digestion speed. Conventional MRI requires the participants to lie down in the magnet bore. Whilst this may have affected bolus spatial position due to gravity, the participants were in the upright position until the relatively quick positioning in the MRI scanner and the data acquisition time; therefore, the effect of position on food boluses’ number and volumes would have been minimal. The same reasoning would apply to the gastric emptying times. It is also important to consider the limitations of the methodology due to the resolution. Whole body MRI imaging is necessarily limited to millimetric resolution and we included here food boluses that could be identified and drawn around on the images. Thus, smaller particles within the stomach could not be identified and single-voxel pixels were not included. The two MRI scanners used for these studies both had a 1.5 T (Tesla) field strength; therefore, the relaxation times of the various components of the meals were comparable. Some differences in imaging sequences and particularly in parameters such as slice thickness (with Meal 3 having been imaged with half the slice thickness used for Meal 1 and Meal 2) could have introduced some bias in the data and, for future studies, it would be recommended to use more comparable set up between sites. The analysis of the images recorded after swallowing allowed us to obtain the best images to investigate the boluses at the time they reach the stomach. Sham-feeding models provide good information on food boluses during the oral step, but they cannot include the swallowing and oesophageal transit steps. As such, the MRI data may also help evaluate the relevance of sham-feeding models and fill this knowledge gap. To the best of our knowledge, this is the first time MRI has been used for these purposes and the results obtained are useful for the improvement of future in vitro and in vivo research. The determination of bolus size within the stomach may also improve understanding of the fate of foods in the stomach of patients with gastroparesis, which is currently performed by scintigraphy [35] with the limitations already described. MRI could also be used in uncovering the difficulties faced by dysphagia sufferers, who have problems with swallowing, related to the oral processing of food, which has implications on bolus processing in the stomach.

## 5. Conclusions

Analysis of MRI images of the stomach from healthy participants after ingesting three different types of meals has shown the value of this technique for the study of bolus volume in the initial moments of gastric digestion. Endpoints such as the number of boluses in the stomach, the percentage ratio of the boluses, the surface area of the boluses and the energy content per mL of bolus could be measured. These new in vivo data from undisturbed organ imaging will improve knowledge of the digestion process, which will, in turn, inform in vitro and in silico modelling of digestion, thus improving their in vitro/in vivo relevance.

The analysis of MRI recorded after ingestion of three different meals revealed significant differences among the boluses found in the stomach in the first few minutes after the meal was consumed. The cohesiveness and resilience of the boluses seem to be the driving factors that lead to the resulting boluses that are bigger than the bolus volume that triggers swallowing. Moreover, the bolus composition also plays its role in the gastric bolus volume. Boluses high in carbohydrate are hydrated, resulting in boluses that represent a high percentage ratio in the stomach but low energy content. Further work is needed to unveil the effect of food composition in the properties of the resulting boluses.

## Figures and Tables

**Figure 1 nutrients-13-03626-f001:**
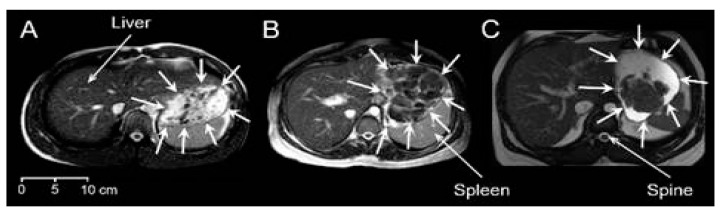
Example axial MRI images for the three different meals. (**A**) Meal 1: chicken and vegetables. (**B**) Meal 2: bread. (**C**) Meal 3: cheese and yogurt. The liver, spleen and spine anatomical landmarks are indicated, and the stomach boundaries are indicated by surrounding white arrows. In this type of MRI image weighting, fluids in the stomach appear bright and solids appear dark.

**Figure 2 nutrients-13-03626-f002:**
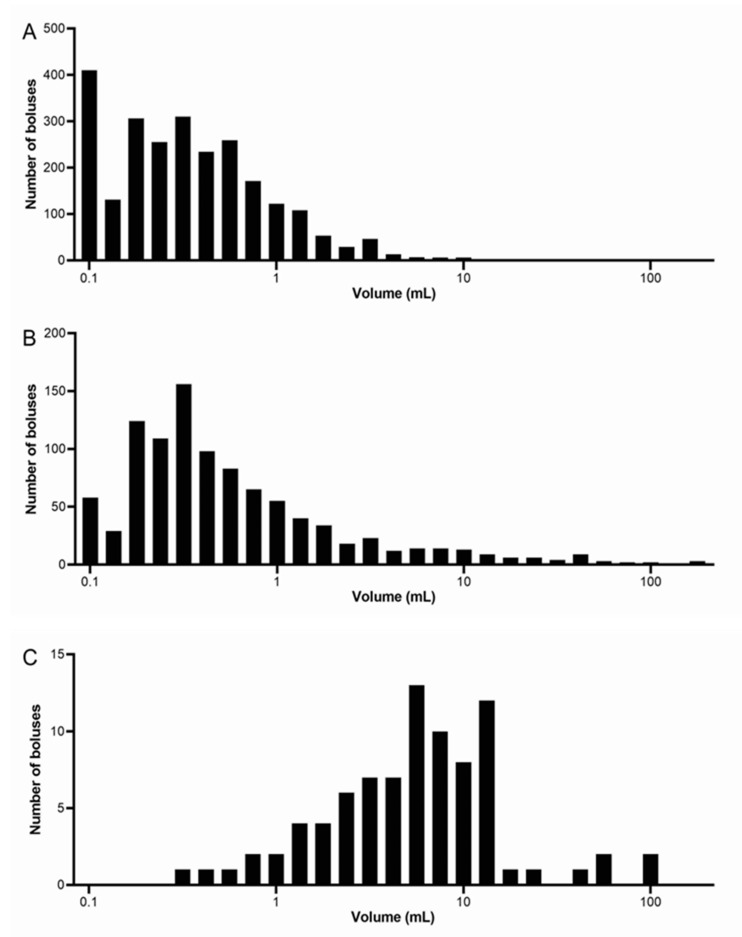
Frequency distribution of the number of food boluses integrated between all nine participants for each meal. (**A**) Meal 1: chicken and vegetables, *n* = 9 healthy participants. (**B**) Meal 2: bread, *n* = 9 healthy participants. (**C**) Meal 3: cheese and yogurt, *n* = 9 healthy participants. Note that different scales were used for each meal on the vertical axes.

**Figure 3 nutrients-13-03626-f003:**
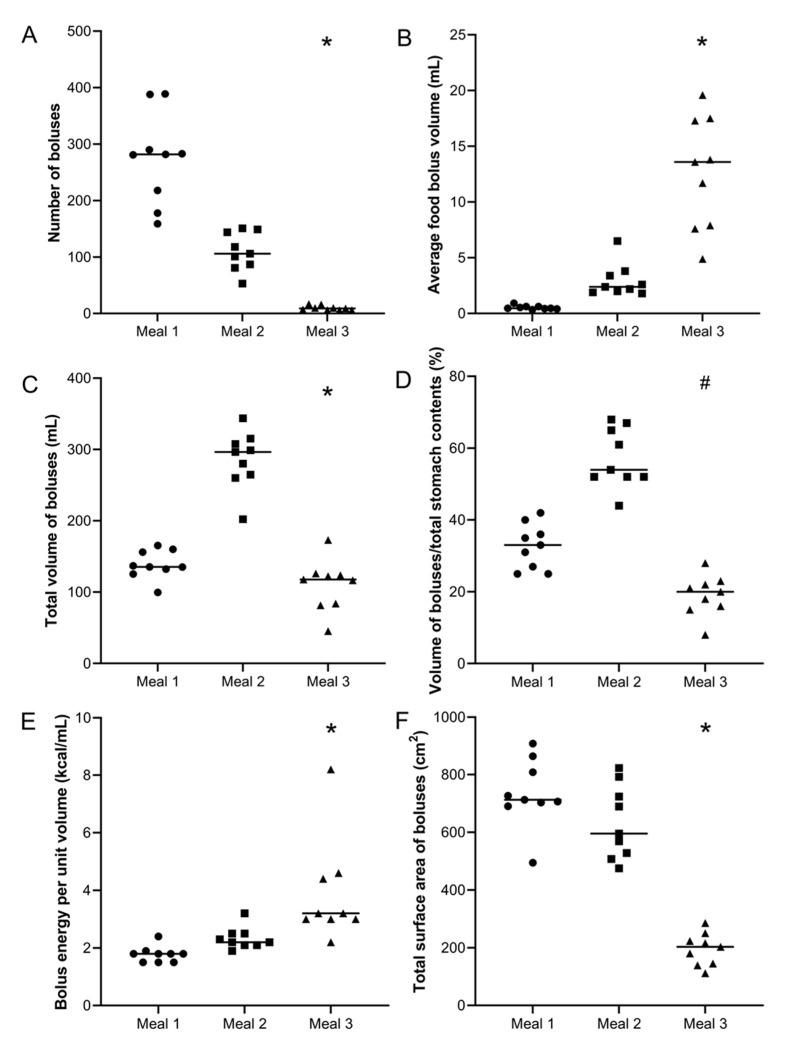
Summary data for each participant and meal. (**A**) Number of food boluses. (**B**) Average food bolus volume. (**C**) Total volume of food boluses. (**D**) Ratio boluses volume/total stomach contents volume (%) and (**E**) Bolus energy per unit volume (kcal/mL). (**F**) Total (cumulative) surface area of boluses. Meal 1: chicken and vegetables, *n* = 9 healthy participants. Meal 2: bread, *n* = 9 healthy participants. Meal 3: cheese and yogurt, *n* = 9 healthy participants. The horizontal lines indicate the median. * *p* < 0.0001 Kruskal–Wallis test; # *p* < 0.0001 1-way ANOVA.

**Figure 4 nutrients-13-03626-f004:**
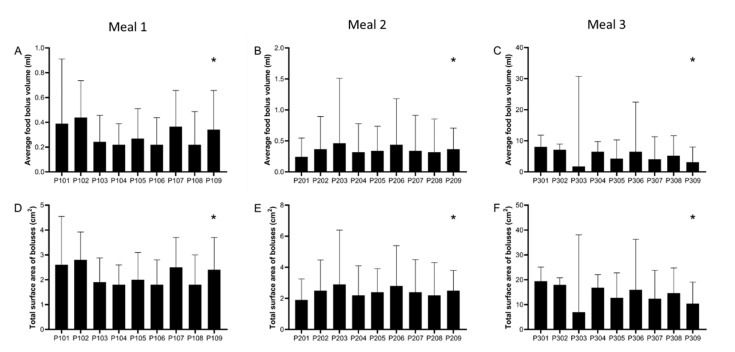
Experimental data (median, interquartile range) from each individual participant and for each meal for: average food bolus volume (**A**–**C**) and total (cumulative) surface area (**D**–**F**) of all food boluses measured in the stomach. Surface area for each bolus was individually calculated from its volume and assuming all boluses were of spherical shape. Meal 1: chicken and vegetables, *n* = 9 healthy participants. Meal 2: bread, *n* = 9 healthy participants. Meal 3: cheese and yogurt, *n* = 9 healthy participants. * *p* < 0.0001 Kruskal–Wallis test.

**Table 1 nutrients-13-03626-t001:** Nutritional composition of the three meals used in the study. Meal 1: chicken and vegetables, Meal 2: bread and Meal 3: cheese and yogurt.

Composition	Meal 1	Meal 2	Meal 3
Energy (kcal)	241	654	373
Fat (g)	12	20	28
Carbohydrates (g)	11	103	6
Fibre (g)	5	5	0
Protein (g)	15	15	25

## Data Availability

The data presented in this study are available in the article’s Appendix A.

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
