# Peer review of "Size and Number of Food Boluses in the Stomach after Eating Different Meals: Magnetic Resonance Imaging Insights in Healthy Humans"

_nutrients, 2021, doi:10.3390/nu13103626_

Round 1

Reviewer 1 Report

All of the prior concerns have been met.

Reviewer 2 Report

No further comments. All remarks have been adequately adressed

This manuscript is a resubmission of an earlier submission. The following is a list of the peer review reports and author responses from that submission.

Round 1

Reviewer 1 Report

Comments on manuscript „Size and number of food boluses in the stomach after eating different meals: magnetic resonance imaging insights in healthy humans“

Decision:  Minor revision

Summary:

The manuscript entitled “Size and number of food boluses in the stomach after eating different meals: magnetic resonance imaging insights in healthy humans” is another interesting work about gastrointestinal processes from the highly experienced MRI workgroup in Nottingham. It is well written and comprehensible. Authors describe the sizes and numbers of food boluses in stomach after intake of vegetables and chicken (meal 1), bread and jam (meal 2) and yoghurt and cheese (meal 3) evaluated from older data sets of already published studies. It turned out that the solid meals led to much smaller and much more solid particles (here called boluses) than the semisolid meal 3.

Unfortunately, the retrospective study design without crossover and/or paired data affects comparability of the results. Comparability between study groups is highly questionable due to high interindividual variability of mastication, salivation and time point of swallowing. A higher sample size would have been adequate for such an unpaired setup, but it is sound that a repetition with a paired setup might be too elaborate. Moreover, several methodical weaknesses were not addressed appropriately. Whether subjects from the original studies also gave written informed consent for the evaluations as presented in the manuscript needs clarification.

But even though there are limitations of study design and methodology, these are interesting results, worth to be published in such a reputable scientific journal like Nutrients after thorough revision.

Broad comments:

To show importance of the work, introduction could need some more information why the food boluses are interesting for other processes and what the influence of their different properties should be. For what does this knowledge about bolus distribution help? (attraction of the reader)

For me the discussion of limitations is lacking. One limitation are the different subjects of the studies. In study 3 males are highly overrepresented whereas they are underrepresented in the other two studies. Males typically have different eating and chewing behaviour compared to women. Moreover, these males have been much older and with higher BMI than subjects in the other studies. The high interindividual variability can cause differences between subject groups in different studies. More limitations are highlighted under specific comments.

Such limitations need to be adequately and transparently addressed.

Nonetheless, I would agree that due to the dramatic differences in size and amount of boluses statistics might still be robust enough.

Furthermore, MRI gives no information about cohesiveness and mechanical properties of boluses, what brings out the question what a “bolus” according to the authors opinion is. Is it every single particle (e.g. meal 1) or more a cohesive mass (meal 3)? Typically, a bolus is the masticated mass of food that is swallowed in one swallow. But this is not what was measured (obvious, can also be seen by the volumes far away from a typical swallowing volume, boluses of solid food like vegetables and chicken tend to disintegrate to smaller particles after reaching the stomach, others tend to form cohesive masses out of several boluses (e.g. bread, semisolids). The term “bolus” needs to be defined better in the introduction. For me, the semisolid masses of meal 3 are not boluses in the established meaning of the word. It seems as if more effort was put into the wording of the conclusion as here authors differentiate between gastric boluses and swallowed boluses.

Meal 2 was administered with much less fluid, that could affect identification of bolus/fluid borders as there is less fluid between the boluses. Thus, they could appear smaller in numbers and larger in size. Please discuss according to obtained results. More remarks on identification of boluses can be found under specific comments.

One needs to be aware, that manual (subjective) selection and calculation of ten of thousands of bolus slices must have been an enormous amount of work. The effort is highly respected.

I believe that for the aimed goal, those data can be evaluated robustly enough in the described manner, but some methodical issues need to be clarified and/or discussed (see specific comments).

Specific comments:

l.21:                             For clarity of abstract: please add “bread and jam”

ll.22-24:                       Due to non-gaussian distribution authors say that medians and IQR were used (see chapter 2.6). This should also be done in the abstract.

ll.75-76:                       What size allows meal components to pass to the duodenum?

ll.81-83:                       is that the opinion of the authors, or is there literature proofing the statement, that bolus size changes during esophageal transit? Please clarify in manuscript.

Chapter 2.1:                please clarify how many subjects had been included to the original studies and from how many subjects the data sets were thus "randomly" selected. How was randomization performed? This might be an important factor to avoid selection bias. Were biometric data of selected subjects representative for the whole subject groups from which they have been randomly selected? Subject groups are not ideally comparable (gender, age, BMI) between the studies. Please discuss.

Chapter 2.2:                To assure comparability, please give masses of whole meals (portion size), which is obviously not the sum of caloric components, as different masses (portion sizes) could lead to different number of boluses. Please also discuss that effect subsequently in discussion part.

Chapter 2.4.                Please give a table including the sequence parameters for imaging of meal 1, 2 and 3. This would be easier and clearer for the reader. For meal 1 and 2 the FOV is missing. Resolution in terms of slice thickness in study 3 is twice as much as in the other studies. Could that increase in resolution account at least partly to the obtained results? Was there a gap between slices? Please discuss how comparable imaging procedures have been, not only in terms of resolution and contrast but also in terms of SNR.

                                    Was meal intake somehow standardized (e.g. with respect to time)? At what time point after meal intake was imaging performed? This might affect comparability of studies, since boluses change during digestion. If timepoints differed this also needs to be discussed.

  1. 159-160: How much darker needed the bolus to be? How was that defined reproducibly? As a percentage of water SI? And if, what about meal 2, where no water was given? Or was it only subjective decision of observer? Was contrast ratio standardized? Are authors sure - especially for meal 1 - that also very watery vegetables (e.g. zucchini) could be identified due to contrast? Where any phantom measurements (e.g. in vitro) performed to assure selectability according to contrast of food components against fluids? Is a discrimination between gouda and yoghurt even possible so that solid cheese particles can be selected in semisolid yoghurt? Images suggest that both mainly appear as large homogeneous mass. Whether this is the meaning of term “bolus” is discussable. Limitations of procedures should be extended and better discussed at ll.396-404.

                                    Please give in manuscript if all evaluations were done by the same observer. How trained was the observer? If not, please clarify and discuss interobserver variability. Where all data sets evaluated randomized or in a specific order? Due the large amount of slices, a training effect of the observer (if only one) could bias the (subjective) results if data sets of meal 1, 2 an 3 where evaluated one after another, especially if the observer was not very experienced at the beginning.

l.169:                           How was surface area calculated? (on page 9, line 290 information is given, but I think this also belongs to methods section)

Obviously boluses were not perfect spheres, as authors state themselves in chapter 3.1. This assumption is probably too far away from reality. From my point of view, the usefulness of these surface data is questionable and the robustness of the calculated values is also questionable even more. I would advise to exclude the surface data from the manuscript, although I see its worth for argumentation of digestion speed. If data should stay in manuscript the high inaccuracy (also due to the limitations given in ll.396-398) needs to be transparently addressed.

Chapter 2.6                 It is clear that authors used non-parametric tests or parametric tests when all three data sets to be compared were normally distributed or not. Please clarify shortly what was done when only one of the data sets could not be proven to be normally distributed. Was non-parametric testing performed even when only one of three data sets did not matched gaussian criteria of Shapiro-Wilk test?

                                    The ANOVAs did not give information where the significant difference was in between the groups. Why was no post-testing performed? (I think GraphPad is does not have post tests for unpaired data)

Figure 2                      Maybe one could somehow highlight the different scale of y-axis in the caption, so that differences are more obvious to the reader? Or a percentaged scaling (size distribution in %) would improve that. This could be added to the figure just right to it, so that Figure 2 consists of six graphs (two for each meal)

Figure 3F:                   Maybe it would be better to name it “cumulative surface area of boluses”? (for me it would be clearer but it is just a suggestion)

Figure 4:                     Mean and SD are given, but it should be median and IQR, as data are obviously not normally distributed (Kruskal-Wallis ANOVA). This would also fit better to descriptions in methods section. Moreover, I would rather present data on bolus numbers than on surface areas. I would exclude surface areas, as I cannot see a benefit from these (biased) data here for interindividual variability.

ll.336-338:                   I do not really agree, that meal 1 and 2 would be “digested” quicker even though the more dispersed particles and higher contact surfaces to gastric media do indeed help for sufficient digestion. (indeed quicker digestion according to particle size can be assumed if meals would be composed comparable) Authors neglect the different caloric densities of the meals and their mechanical properties. Moreover, what is meant with “digested”? Only chemical breakdown, or volume transfer to small intestine, or calory transfer to small intestine? Or also physical breakdown (antral mill) which is probably much slower for the solids? Even the higher surface could not be sufficient for faster chemical breakdown compared to a semisolid meal, which is better permeable for the gastric fluids and in which gastric fluids can be mixed into. Please discuss.

To bring such a statement, gastric emptying need to be taken into account. In contrast to the aforementioned statement, Meal 3 is indeed emptied slower than meal 1 and 2 (according to the authors own literature)

ll.343-346:                   According to the cited literature of Siegel et al. there is of course a lag phase, but that does not mean that there was no gastric emptying during this phase, it is just slower. Thus, it is wrong to say “that gastric emptying would not have started.” in “the time in which the images of meal 1 were taken” Taken into account the cited literature, gastric emptying already taken part was probably even more pronounced for meal 2 and 3?

ll.345-346:                   At which time point were images taken relative to the different meal intakes? Please discuss comparability of study setups accordingly.

l.366:                           Similar composition in what terms? Physical properties are completely different and caloric values also differ. Please clarify.

ll.375-381:                   Caloric density (kcal/mL or kcal/g) according to portion sizes need to be taken into account. That is one of the main factors for gastric emptying. Depending on how half emptying time (time point on which half of the volume is emptied?) is calculated it is heavily biased by portion size and does not reflect gastric emptying rates. Please clarify.

l.395:                           Maybe it would be better to say “Nonetheless, this work shows that initial size of …” as the aforementioned statements on meal 1 and meal 3 indicate opposed effects of bolus size.

ll.401-402:                   If one starts quantifying boluses in the size range of single voxels it is likely to include darker voxels due to noise into to the calculations. This might be especially the case for the two older studies of meal 1 and 2 in which much more smaller boluses were measured. Please discuss.

ll.424-425:                   It is misleading to say that images were taken minutes after swallowing of boluses. They were probably taken minutes (so far undefined time span in manuscript!) after completion of the meal. That is a difference as there is time for processing and changes especially for the first boluses.

l.443:                           Subjects gave written informed consent for the three original studies and their aims, but probably intragastric bolus evaluation was not part of the subject information at this time point. Authors need to specify whether subjects also gave written informed consent for this evaluation. Otherwise the presented data were not allowed not be acquired nor to be published. Authors might need to share their subjects information forms so that can be proven (publication e.g. in supplementary files is not necessary to my point of view, it is only to prove ethical correct work in reviewing process)

Reviewer 2 Report

The methodology is appropriate and described well. Results are presented clearly. The purpose of this secondary analysis appears to be determine whether MRI imaging would allow visualization of food boluses and to provide knowledge on food bolus size. The discussion notes the advantages of MRI technology and the limitations related to the ability to determine the size of the boluses. Also consider that the MRI imaging has the subject supine which may affect gastric emptying times. The information related to food boluses, effects of meal content, and liquid is missing some key references related to these topics. Consider the limitation of differences in mastication among individuals as well. 

While this is an interesting study, the discussion and conclusions could be stronger. It is not clear how these findings could or should inform practice of why it is important to others.